ALKBH5 modulation of ferroptosis in recurrent miscarriage: implications in cytotrophoblast dysfunction

Qin Chuanmei 1
Wu Jiayi 2 3 4
Wei Xiaowei 1
Liu Xueqing 2 3 4
Lin Yi yilinonline@sjtu.edu.cn 1
1 Shanghai Jiao Tong University School of Medicine Affiliated Sixth People’s Hospital, School of Medicine, Shanghai Jiao Tong University , Shanghai , China
2 The International Peace Maternity and Child Health Hospital, School of Medicine, Shanghai Jiao Tong University , Shanghai , China
3 Shanghai Key Laboratory of Embryo Original Diseases , Shanghai , China
4 Institute of Birth Defects and Rare Diseases, School of Medicine, Shanghai Jiao Tong University , Shanghai , China
Mokhtar Mohd Helmy
Electronic publication date: 2024 Oct 18
Publication date: 2024
Volume: 12
Electronic Location ID: e18227
Received 2024 Jan 10; Accepted 2024 Sep 13
Copyright: ©2024 Qin et al.
Copyright year: 2024
Copyright holder: Qin et al.
License: This is an open access article distributed under the terms of the Creative Commons Attribution License, which permits unrestricted use, distribution, reproduction and adaptation in any medium and for any purpose provided that it is properly attributed. For attribution, the original author(s), title, publication source (PeerJ) and either DOI or URL of the article must be cited.
License URL: https://creativecommons.org/licenses/by/4.0/

Keywords: ALKBH5, FTL, Cytotrophoblast, Ferroptosis, Recurrent miscarriage

Funding: The National Natural Science Foundation of China No. 82471722 No. 82171669 No. 81971403 The Shanghai Jiao Tong University Trans-Med Awards Research No. 20210201 The Funds for Outstanding Newcomers, Shanghai Sixth People’s Hospital No. X-3664 This work was supported by the National Natural Science Foundation of China (No. 82471722, No. 82171669, No. 81971403), the Shanghai Jiao Tong University Trans-Med Awards Research (No. 20210201), and the Funds for Outstanding Newcomers, Shanghai Sixth People’s Hospital (No. X-3664). The funders had no role in study design, data collection and analysis, decision to publish, or preparation of the manuscript.

==============================
Background

As one of the most common and abundant internal modifications of eukaryotic mRNA, N6-methyladenosine (m6A) modifications are closely related to placental development. Ferroptosis is a newly discovered form of programmed cell death. During placental development, placental trophoblasts are susceptible to ferroptosis. However, the interactions of m6A and ferroptosis in trophoblast physiology and injury are unclear.

Methods

Recurrent miscarriage (RM) was selected as the main gestational disease in this study. Published data (GSE76862) were used to analyze the gene expression profiles in patients with RM. The extent of m6A modification in total RNA of villous tissues between patients with RM and healthy controls (HC) was compared. ALKBH5 (encoding AlkB homolog 5, RNA demethylase) was selected as the candidate gene for further research. Quantitative real-time reverse transcription PCR, western blotting, and immunohistochemistry (IHC) confirmed the elevated expression of ALKBH5 in the cytotrophoblasts of patients with RM. Then, cell counting kit-8 assays, glutathione disulfide/glutathione quantification, 2′,7′-dichlorfluorescein-diacetate staining, and malonaldehyde assays were used to explore the alterations of ferroptosis-related characteristics following RAS-selective lethal (RSL3) stimulation after overexpression of ALKBH5. Thereafter, we re-analyzed the published RNA sequencing data upon knockdown of ALKBH5, combined with published tissue RNA-seq data, and FTL (encoding ferritin light chain) was identified as the ferroptosis-related gene in cytotrophoblasts of patients with RM that is regulated by ALKBH5. Finally, western blotting and IHC confirmed the increased expression of FTL in the cytotrophoblasts from patients with RM.

Results

Total m6A levels were decreased in patients with RM. The most significant differentially m6A-related gene was ALKBH5, which was increased in patients with RM. In vitro cell experiments showed that treatment with RSL3 resulted in increased cell death and upregulated ALKBH5 expression. Overexpression of ALKBH5 alleviated RSL3-induced HTR8 cell death and caused decreased levels of intracellular oxidation products. Published transcriptome sequencing revealed that FTL was the major ferroptosis-related gene regulated by ALKBH5 in the villous tissues of patients with RM. Consistent with the expression of ALKBH5, FTL was increased by RSL3-induction and increased in patients with RM.

Conclusion

Elevated ALKBH5 alleviated RSL3-induced cytotrophoblast cell death by promoting the expression of FTL in patients with RM. Our results supported the view that ALKBH5 is an important regulator of the ferroptosis-related etiology of RM and suggested that ALKBH5 could be responsible for epigenetic aberrations in RM pathogenesis.

Introduction

Recurrent miscarriage (RM) is defined as two or more consecutive spontaneous abortions with the same sexual partner after confirmed intrauterine pregnancies and before 20–24 gestational weeks (Dimitriadis et al., 2020). Recurrent miscarriage occurs in approximately 1–2% of all couples trying to conceive (Bender Atik et al., 2023; Quenby et al., 2023). The cause of RM is complex, including chromosomal errors, uterine malformation, and autoimmune dysfunction; however, the etiology of 40–50% of cases remains unknown (Daimon et al., 2020; Dimitriadis et al., 2020).

Maintenance of pregnancy is a complex and highly regulated biological process requiring coordination of nutrients and immunity between the mother and the fetus. Moreover, this coordination requires a functional placenta. The placenta, mainly consisting of maternal decidua and fetal villi, plays a critical role throughout the process of pregnancy (Centurione et al., 2018). As soon as the implantation of the embryo begins, development of the placenta is initiated by the generation of trophoblast cells from the trophectoderm of the blastocyst (Turco et al., 2018). Cytotrophoblasts (CTBs), an inner layer of the chorionic villi, can differentiate into extravillous CTBs or fuse to form the external layer of syncytiotrophoblasts (STBs) (Costello & Fisher, 2021; Hemberger, Hanna & Dean, 2020). Before gestational week 11, the development of the conceptus relies on uterine secretions; however, thereafter, a villous placenta-blood interface is effectively functional, and the exchange of nutrients, gases, and metabolic waste products mainly relies on contact between the maternal blood and fetal villi (Burton et al., 2002; Erlich et al., 2019; Jones, Choudhury & Aplin, 2015). Placental dysfunction associated with impaired trophoblast function might lead to miscarriage, fetal growth restriction, preeclampsia, and stillbirth (Brosens et al., 2011; Turco et al., 2018).

Epigenetic modifications, including histone modifications, DNA methylation, non-coding RNAs, and RNA methylation, play functional roles in maternal-fetal medicine (Hocher & Hocher, 2018; Wu et al., 2023). N6-methyladenosine (m6A) is the most common epigenetic RNA modification, and was reported to modulate the biological process of placenta formation and development (Wu et al., 2023). The m6A modification is a complex and reversible process coregulated by m6A writers, erasers, and readers (Zhao, Roundtree & He, 2017). It has been reported the m6A writer complex mainly consists of methyltransferase-like 3 (METTL3), methyltransferase-like 14 (METTL14), and Wilms tumor 1-associated protein (WTAP), while the m6A demethylase (erasers) include fat mass and obesity-associated (FTO) and alkB homolog 5 (ALKBH5). The m6A readers mainly comprise the YT521-B homology domain protein family (YTHDF) (including YTHDF1-3) (Zhang et al., 2023). It has been reported that m6A dysregulation might lead to gestational diseases, such as RM, preeclampsia, and gestational diabetes mellitus (GDM) (Li et al., 2019; Taniguchi et al., 2020; Wang et al., 2021). Increased ALKBH5 was reported to inhibit trophoblast invasion at the maternal-fetal interface of patients with RM (Li et al., 2019). Under fear stress, m6A modifications have been reported play an important role in placental dysfunction during pregnancy (Wang et al., 2022). Moreover, downregulation of m6A is involved in the development of GDM (Wang et al., 2021).

Ferroptosis is a programmed cell death dependent on iron (Jiang, Stockwell & Conrad, 2021). Early in placental development, trophoblast cells experience physiological hypoxia, which is intimately linked to ferroptosis, leading to placental dysfunction and reproductive disorders (Beharier, Kajiwara & Sadovsky, 2021; Ng, Norwitz & Norwitz, 2019). The small molecule, RAS-selective lethal (RSL3), is one of the most important and universal ferroptosis inducers (Yang & Stockwell, 2016). It has been reported that Jianwei Shoutai Pill (JSP) and Ferrostatin-1 (Fer-1) could reverse the impairment of HTR8 cell (a trophoblast cell line) viability and migration induced by RSL3. Moreover, in a mouse model, JSP and Fer-1 decreased the embryo resorption rates of RM mice (CBA/J × DBA/2 mice) (Lai et al., 2024). This suggests that ferroptosis might cause adverse pregnancy outcomes.

The goal of the present study was to clarify the link between m6A, ferroptosis, and RM pathogenesis. We focused on the functional m6A and ferroptosis-related genes in RM, including transcriptomic analysis and identifying the mechanisms affecting the function of trophoblasts.

Materials & Methods

Patient characteristics

Fifteen healthy controls (HC) (mean age, 30.40 ± 2.50 years; mean gestation week, 7.59 ± 0.81 weeks) and 15 patients with RM (mean age, 31.20 ± 4.18 years; mean gestation week, 7.64 ± 1.41 weeks) were recruited from the International Peace Maternity & Child Health Hospital from September 2020 to June 2021. Some samples contained only a small amount of villous tissue; therefore, it was not possible to perform multiple experiments simultaneously. We used 15 samples from the HC group and 15 samples from the RM group in the quantitative real-time reverse transcription PCR (qRT-PCR) experiment. In the western blotting (WB) experiment, we used 14 samples from each group. In the immunohistochemistry (IHC) experiment, we used 10 samples from each group. All recruited individuals with HC or RM were under 35 years old and terminated at weeks 6–11 of gestation. Chromosomal abnormalities, endocrine disorders, endometritis, abnormal uterine structure, cervical insufficiency, and other identified etiologies of RM were excluded. The RM group was diagnosed with embryonic arrest and underwent artificial abortion to remove intrauterine tissue. Meanwhile, the HC group has had a history of successful childbirth and chose artificial abortion to terminate an unwanted pregnancy. The study was approved by the Medical Ethics Committee of the International Peace Maternity & Child Health Hospital of China Welfare Institute, Shanghai [approval number (GKLW) 2021-49]. Written informed consent was obtained from all participants (HC, RM).

Tissue collection

All villous tissues were collected immediately after artificial abortion and cleaned using phosphate-buffered saline (PBS) (Hyclone, Logan, UT, USA) as described previously (Wei et al., 2022). For qRT-PCR, part of the villous tissue was stored in RNAlater (Thermo Fisher Scientific, Waltham, MA, USA) overnight at 4 °C. Samples were then transferred into liquid nitrogen for long-term storage. For western blotting, part of the villous tissue was snap frozen in liquid nitrogen within 20 min after sampling. For immunohistochemistry, part of the villous tissue was fixed using 4% paraformaldehyde (PFA) (BBI Life Sciences, Shanghai, China) for 24 h, and then embedded in paraffin.

Prussian blue staining

An Iron Stain Kit (ab150674; Abcam, Cambridge, UK) was used to carry out iron staining in tissue sections. Tissue sections on slides were deparaffinized, rehydrated, and then incubated in working iron stain solution for 3 min. After rinsing with distilled water, the sections were stained with nuclear fast red solution for 5 min and rinsed with distilled water four times. The sections were dehydrated in 95% alcohol followed by absolute alcohol and mounted. Blue staining showed the non-chelated iron in the placental villi.

Cell culture

The HTR-8/SVneo cell line was a kind gift from Dr. P.K. Lala (University of Western Ontario, London, ON, Canada) (Graham et al., 1993), which was cultured in DMEM/F12 (Gibco, Grand Island, NY, USA) plus 10% fetal bovine serum (Yeasen, Shanghai, China) and 1% penicillin/streptomycin antibiotics (Gibco, Grand Island, NY, USA). Cells were cultured in a 37 °C cell culture incubator under a 5% CO2 atmosphere.

Transfection protocol

The non-targeting control small interfering RNA (siRNA) (siNC) and ALKBH5 siRNA (siALKBH5) were purchased form GenePharma (Shanghai, China). Oligofectamine® 2000 Reagent (Invitrogen, Carlsbad, CA, USA) was used for the transfection of siRNA according to the manufacturer’s instructions. For overexpression, the pLV-hALKBH5 plasmid and the corresponding pLV empty control plasmid (Vector) were purchased from Cyagen company (Suzhou, China). Transient transfection was performed using the JetPRIME® transfection reagent (Polyplus transfection™; JetPRIME, Illkirch, France). Using a 6-well plate as an example, 2 µg of plasmid was added in 200 µl jetPRIME® buffer, followed by the addition of 4 µl of jetPRIME® reagent, mixed and incubated for 10 min at room temperature. Then, the mixture was added to 2 ml of serum-containing medium in the 6-well plate. Cells were transfected when they reached 30–40% confluence. RNA and protein were collected at 48 h after transfection. Experiments were repeated three times.

RNA isolation and qRT-PCR

The villous tissues or cells were lysed using Trizol (Life Technologies, Carlsbad, CA, USA) and total RNA was extracted according to a reported method (Rio et al., 2010). One microgram of RNA was reverse transcribed to cDNA using Evo M-MLV RT Master Mix (Accurate Biology, Hangzhou, China). Quantitative real-time PCR (qPCR) was performed using the cDNA as the template and the SYBR Green Premix Pro Taq HS qPCR Kit (Accurate Biology), together with the primers listed in Table S1. Relative mRNA expression was normalized to ACTB (encoding β-actin) expression, calculated using the 2−ΔΔCt method in cells, while the 2−ΔCt method was used in human tissue samples (Livak & Schmittgen, 2001; Schmittgen & Livak, 2008). More details are available in the supplemental data: MIQE checklist.

Western blotting

Tissues and cells were lysed in radioimmunoprecipitation assay buffer (Thermo Fisher Scientific) containing a protease inhibitor cocktail (Solarbio, Beijing, China). The protein concentration was measured using a bicinchoninic acid (BCA) Protein Assay (Thermo Fisher Scientific). Then, the proteins were denatured by heating at 100 °C for 10 min. Next, 10 µg of protein was loaded into each well and separated using sodium dodecyl sulfate-polyacrylamide gel electrophoresis (SDS-PAGE). After electrophoresis, the proteins were transferred onto 0.2-µm polyvinylidene fluoride (PVDF) membranes (Millipore, Milford, CT, USA). The membrane was blocked with 5% non-fat skim milk at room temperature for 1 h and then incubated with primary antibodies at 4 °C overnight. The following day, the membranes were incubated with secondary antibodies conjugated with horseradish peroxidase (HRP) for 1 h at room temperature. The antibodies and dilutions used were as follows: anti-ALKBH5 (1:1000, ab195377; Abcam, Cambridge, UK), anti-ferritin light chain (FTL) (1:1000, ab75973, Abcam), anti-β-actin (1:10000, 66009-1-Ig; Proteintech, Rosemont, IL, USA), Goat Anti-Rabbit IgG antibody (1:5000, ab288151; Abcam), and Goat Anti-Mouse IgG antibody (1:5000, ab7063; Abcam). Signals were detected using a Chemiluminescent HRP Substrate (Millipore) and analyzed using Image J software (NIH, Bethesda, MD, USA). Images of uncropped western blots can be found in Fig. S1.

Quantification of m6A in total RNA

Total RNA methylation was quantified using a EpiQuik M6A RNA Methylation Quantification Kit (Epigentek, Farmingdale, NY, USA). Total RNA was extracted using Trizol and diluted to a final concentration of 200 ng/µl. The amount of RNA used per sample was 200 ng. The RNA samples and the diluted positive control were transferred to corresponding wells in a binding solution-treated 96-well plate. Each sample was analyzed using two technical replicates. After binding of RNA, the 96-well plate was incubated with the capture antibody at room temperature for 1 h, and then washed four times. Subsequently, the detection antibody was added to the wells of the plate and incubated for 30 min. After rinsing with washing buffer for five times, the plate was incubated with developer solution for about 5 min. Stop solution was added to halt the enzyme reaction after the color turned blue. The plate was then read for absorbance at 450 nm. The relative m6A RNA methylation status was calculated according to the absorbance. First, we drew a standard curve based on the absorbance of positive controls at different concentrations and determined the slope of the standard curve using linear regression. m6A (ng) = (Sample absorbance –blank absorbance)/slope. To demonstrate the changes in the experimental group compared with the control group more clearly, we used the relative m6A value to display the results. Taking the relative m6A levels in villous tissue as an example: relative m6A = Sample absorbance/[sum(HC group)/15].

Immunohistochemistry (IHC) assay

Villous tissues from 10 patients with RM and 10 HC controls were washed with PBS immediately after collection and fixed in 4% paraformaldehyde. After 48 h, the tissues were paraffin-embedded and serially sectioned. A mouse and rabbit specific HRP/diaminobenzidine (DAB) (ABC) Detection IHC Kit (ab64264, Abcam, Waltham, MA, USA) was used for IHC. The assay was performed according to the manufacturer’s instructions. After dewaxing and rehydration, slices were incubated with primary antibodies against ALKBH5 (1:1000, ab195377; Abcam) (Wang et al., 2023), FTL (1:500, ab69090, Abcam, UK) (Raha et al., 2022), or Rabbit IgG (1:500, #3900, Cell Signaling Technology, Danvers, MA, USA) (Tong et al., 2023) diluted in primary antibody dilution (P0277; Beyotime, Haimen, China). All antibodies were commercially acquired and have been validated for their specificity in previous studies. Negative controls were performed during pre-experiments. The following day, slides were incubated with the biotinylated secondary antibody, stained with diaminobenzidine (DAB), and counterstained with hematoxylin. After dehydration and mounting, slides were observed and photographed under a microscope. Scoring of slides was performed independently by two pathologists in a blinded manner according to the published literature (Liu et al., 2020; Zhao et al., 2019).

Immunofluorescent staining

For immunofluorescence, HTR8 cells were seeded in 24-well plates 1 day before the experiment. The next day, at a confluence of 40%, RSL3 was added to the plates at different concentrations (0 µM, 2.5 µM, 5 µM). After 24 h, the cells were fixed using 4% paraformaldehyde for 15 min at room temperature, blocked and permeabilized with immunol staining blocking buffer (P0102; Beyotime) for 20 min at room temperature, and then incubated with primary antibodies against ALKBH5 (1:500, 16837-1-AP, Proteintech) overnight at 4 °C (Liu et al., 2021). Next day, the cells were incubated with Alex Fluor 488-conjugated goat anti-Rabbit IgG (Life Technologies, Carlsbad, CA, USA). After washing with PBS three times, the cells were mounted with mounting medium with 4,6-diamino-2-phenylindole (DAPI) (ab104139, Abcam) and observed under a fluorescence microscope (Leica DMi8 microscope; Leica Microsystems, Wetzlar, Germany).

Cell Counting Kit-8 (CCK8) assay

About 2500 cells/per well were plated in 96-well plates 1-day before drug treatment. At 1 h before drug treatment, corresponding volumes of dimethyl sulfoxide (DMSO; D2438, Sigma, St. Louis, MO, USA), Fer-1 (1 µM, S7243, Selleck, Houston, TX, USA), UAMC-3203 (2 µM, S8792, Selleck), necrosulfonamide (1 µM, S8251, Selleck), carbobenzoxy-valyl-alanyl-aspartyl-[O-methyl]-fluoromethylketone (Z-VAD-FMK; 10 µM, S7023, Selleck), Hydroxychloroquine (HCQ) Sulfate (1 µM, S4430, Selleck) were added to cell culture at 2 h before the addition of RSL3 (0, 1, 2, 3, 4, 5, 6, 7, and 8 µM), respectively. Then, 24 h later, the optical density at 450 nm (OD450) was assessed using a Cell Counting Kit-8 (Yeasen).

To clarify the role of ALKBH5 in RSL3-induced cell death, about 2000 cells/per well were plated in 96-well plates 1 day before transfection of plasmids, at which point they had reached ∼30% confluence. At 1 day after transfection, the cells were treated with RSL3 (5 µM) (Selleck Chemicals, Munich, Germany) or equal volumes of solvent (DMSO) with three duplicate wells in each group. After 24 h, the OD450 was assessed.

Glutathione disulfide/glutathione (GSSG/GSH) quantification

Analysis of GSSG/GSH was performed using a GSSG/GSH Quantification kit (G263, Dojindo, Kumamoto, Japan) according to the manufacturer’s instructions. We collected 1 ×107 HTR8 cells and washed them with PBS, which were then lysed twice by freezing and thawing with 80 µl 10 mM HCl. After lysis, 20 µl of 5% sulfosalicylic acid (SSA) was added to cell lysate. The supernatant was collected by centrifugation and formulated in ddH2O to a final concentration of 0.5% SSA. Then, 40 µl of GSSG standard solution, GSH standard solution, GSSG sample solution and GSH sample solution were added to 96-well plates, respectively. Next, 60 µl of buffer solution was added to the plates, which were incubated at 37 °C for 1 h. After incubation, 60 µl of substrate working solution and 60 µl of enzyme/coenzyme working solution were added successively. The plates were then moved to a 37 °C incubator for 10 min and the absorbance was measured at 405 nm.

2′,7′-dichlorfluorescein-diacetate (DCFH-DA) staining

A reactive oxygen species (ROS) assay kit (R252, Dojindo) was used to detect ROS. HTR8 Cells were plated in six-well plates 1 day before transfection. Next day, the pLV empty and pLV-hALKBH5 plasmid were transfected into the cells. After 24 h, about 3,500 cells/per well from each group were plated in 96-well plates, separately. The following day, the HTR8 cells were washed twice with Hank’s balanced salt solution (HBSS). Then, the DCFH-DA working solution was added to the plate and the plate was incubated at 37 °C with 5% CO2 for 30 min. After incubation, the cells were washed twice with HBSS. Corresponding volumes of DMSO and 5 µM RSL3 were added to the cell medium, respectively, to resuspend the cells, and then the plate was then incubated for 2 h. Finally, the cells were washed twice with HBSS and photographed under a fluorescence microscope immediately.

Malonaldehyde (MDA) assay

The MDA assay was performed using a lipid peroxidation MDA Assay Kit (S0131S, Beyotime). HTR8 cells overexpressing ALKBH5 and pLV-transfected control cells were lysed using cell lysis buffer (P0013, Beyotime). Following centrifugation for 10 min at 12, 000 × g, the supernatant was collected, and the protein concentration of each sample was determined using the BCA Protein Assay (Thermo Fisher Scientific) and adjusted to the same concentration with lysis buffer. All the above operations were carried out in an ice bath or at 4 °C. Next, 100 µl of either control, sample, or standard were added to EP tubes. The MDA working solution (200 µl) was then added to each tube, followed by incubation at 100 °C for 15 min. After incubation, the tubes were centrifuged and 200 µl of the supernatant of each sample to be measured was added into a 96-well plate. Then, the absorbance at 532 nm was detected. Finally, the MDA content of each sample was calculated according to the standard curve. The formula obtained for the calculation based on our experimental standard samples was: MDAnmol/mg protein=72.632×Sample absorbance−2.717610×Sample protein content.

Analysis of published RNA-sequencing data

Published data (GSE76862) were analyzed using GEO2R (NCBI). Gene ID conversion, gene ontology (GO), and Kyoto Encyclopedia of Genes and Genomes (KEGG) analyses were carried out using the clusterProfiler package.

To understand the transcriptional changes caused by ALKBH5, published RNA-seq data determined under conditions of ALKBH5 knockdown were re-analyzed (Li et al., 2019). Differential expression between the groups was assessed using the Limma package. Genes were selected according to a significance threshold of fold change (siALKBH5/siNC) ≥ 1.5 or ≤ 0.67 and an adjusted P value <0.05 (Grunseich et al., 2018).

Statistical analysis

All experiments were repeated three times. The results are shown as mean values ± standard deviation. The Shapiro–Wilk test was utilized to assess normality. For data that did not conform to a normal distribution, the Mann–Whitney test was used to analyze the statistical differences. Differences between two groups were compared using Student’s t-test or Welch’s t-test for data displaying a normal distribution. Comparisons between multiple groups were performed using one-way ANOVA with Benjamini–Hochberg false discovery rate correction for data with normal distribution and equal variance. P < 0.05 was considered statistically significant. GraphPad Prism software version 9 (GraphPad Inc., La Jolla, CA, USA) was used for the statistical calculations and presentation.

Results

Iron accumulation in villous trophoblasts of patients with RM

To examine the presence of iron overload in placental villi, we performed Prussian blue staining of villous tissue sections from the HC and RM groups. Prussian blue staining showed iron accumulated at the inner cytotrophoblast layers, and there were significantly more particles in the RM group compared with those in the HC group (Fig. 1A). To identify the role of ferroptosis-related genes in the regulation of the function of villous trophoblasts during the occurrence of RM, we performed an intersection analysis with the published data (GSE76862) and the FerrDB database (Zhou & Bao, 2020). The results identified the top 10 most significant increased and decreased ferroptosis-related genes in the villi of patients with RM (Fig. 1B). Then, we performed GO and KEGG analyses using the 20 genes (Fig. 1C). We also identified the protein–protein interactions with the 20 genes using the STRING database (Fig. 1D). These results suggested that disordered iron metabolism in villous trophoblasts was associated with the onset of RM.

Increased ALKBH5 in cytotrophoblasts of patients with RM

To detect the role of m6A in the regulation of the function of placental villi, we examined the global m6A levels in placental villi of the patients with RM and the HCs. The results showed that the global m6A level was lower in patients with RM compared with that in the HCs (Fig. 2A). Then, we tested the mRNA levels of the major functional m6A-related genes of the trophoblast tissues from patients with RM or the HCs (Fig. 2B). Considering the decreased global m6A levels in the villous tissues of patients with RM, we chose ALKBH5, encoding a m6A demethylase, as the target gene for further research. Subsequent qRT-PCR analysis showed that the ALKBH5 mRNA levels were increased in the villous tissues of patients with RM compared with those in the HCs (Fig. 2C), consistent with the protein levels detected by western blotting (Figs. 2D, 2E). Immunohistochemical staining showed ALKBH5-positive cells in the first-trimester villous tissues, primarily within the cytotrophoblasts. The results also showed much stronger expression of ALKBH5 in the tissues from patients with RM compared with that in the tissues from the HCs (Figs. 2F–2H). Thus, ALKBH5 expression was increased in cytotrophoblasts of patients with RM.

Figure 1 Iron accumulation in villous trophoblasts of patients with RM.

(A) Prussian blue staining on villous tissues from patients with RM (N = 10) or HC (N = 10). (B) Gene lists from published GSE76862 and the FerrDB database ranked by fold change. (C) GO and KEGG enrichment analysis performed using the genes listed in (B). (D) Network analysis of the genes listed in (B) from the String database.

Figure 2 ALKBH5 was increased in the villous tissues from patients with RM.

(A) Total m6A level of each villous tissue sample from patients with RM (n = 15) or HC (n = 15). (B) Detection of m6A reader, writer, and eraser mRNA expression using qRT-PCR. (C) qRT-PCR results showing that ALKBH5 expression was increased in the villous tissues of patients with RM (n = 15) compared with that in HC (N = 15).(D, E) Western blotting results showing that ALKBH5 levels were increased in the villous tissues of patients with RM (n = 14) compared with that in HC (N = 14). (F, G, H) Representative immunohistochemical images depicting the expression of ALKBH5 in the villous tissues from patients with RM (n = 10) and HC (n = 10). (CTB, cytotrophoblast; STB, syncytiotrophoblast; * p < 0.05; ** p < 0.01).

RSL3 increased ALKBH5 during in vitro cell experiments

The trophoblast cell line HTR8/SVneo was used to mimic cytotrophoblasts to clarify the function of ALKBH5 using in vitro experiments. The CCK8 assay showed that RSL3-induced cell death was inhibited by Fer-1 (a ferroptosis inhibitor) and UAMC-3203 (a ferroptosis inhibitor), but not by necrosulfonamide (a necrosis inhibitor), Z-VAD-FMK (an apoptosis inhibitor), or HCQ (an autophagy inhibitor) (Fig. 3A). These results were supported by the corresponding images of HTR8 cells treated as in Figs. 3A and 3B. Quantification of m6A in total RNA showed that m6A levels were decreased by RSL3 treatment (Fig. 3C). Then, qRT-PCR showed that ALKBH5 expression was increased by RSL3 induction (Fig. 3D), which was consistent with the western blotting results (Figs. 3E, 3F) and immunofluorescent staining (Fig. 3G). Considering the mode of cell death induced by RSL3, we suggested that ALKBH5 expression was upregulated during the process of ferroptosis.

Figure 3 RSL3 promoted the expression of ALKBH5.

(A) RSL3-induced HTR8 cell death can be rescued by Ferrostatin-1 (Fer-1) and UAMC-3203, but not by necrosulfonamide, Z-VAD-FMK and Hydroxychloroquine (HCQ) Sulfate (HCQ). When RSL3 is added alone, the average cell viability percentages of 93.95% at 2 µM, 80.02% at 3 µM, and 42.50% at 5 µM. Based on these results, we chose to use concentrations of 0 µM, 2.5 µM, and 5 µM in the upcoming study. (B) Representative images of HTR8 cell morphology after 24 h of treatment as in (A). (C) Quantification of m6A in total RNA after RSL3 induction for 24 h. (D) qRT-PCR showing that ALKBH5 expression was increased after RSL3 induction for 24 h. (E, F) Western blotting results showing that ALKBH5 levels were increased after RSL3 induction for 24 h. (G) Immunofluorescent images showing HTR8 cells with elevated ALKBH5 expression after RSL3 induction for 24 h. Experiments were repeated three times. (** p < 0.01).

ALKBH5 attenuated the events leading to ferroptosis

The characteristics of ferroptosis include insufficient antioxidant capacity (Chen et al., 2023) and enhanced glutathione GSSG/GSH levels (Zhu et al., 2022). Considering the elevated ALKBH5 expression in the cytotrophoblasts of patients with RM, we established HTR8 cells overexpressing ALKBH5 (Figs. S2A–2C). The CCK8 assay showed higher cell viability after ALKBH5 overexpression following RSL3 treatment (Fig. 4A). Consistent with the results for cell viability, the GSSG/GSH ratio decreased after ALKBH 5 overexpression (Fig. 4B). Moreover, ALKBH5 decreased the ROS levels induced by RSL3, as detected using the DCFH-DA fluorescent dye (Fig. 4C). Furthermore, the MDA levels in cells overexpressing ALKBH5 were significantly decreased compared with those in the control group (Fig. 4D). Taken together, these results indicated that overexpression of ALKBH5 attenuated the characteristics of ferroptosis.

Figure 4 Characteristics of ferroptosis of HTR8 cells after overexpression of ALKBH5.

(A) CCK8 assay results showing that overexpression of ALKBH5 alleviated RSL3-induced cell death. (B) GSSG/GSH quantification showing that overexpression of ALKBH5 alleviated RSL3-induced GSSG/GSH imbalances. (C) DCFH-DA staining showing that overexpression of ALKBH5 alleviated RSL3-induced generation of ROS. (D) MDA assay showing that overexpression of ALKBH5 alleviated RSL3-induced generation of MDA. Experiments were repeated three times. (* p < 0.05; ** p < 0.01).

FTL is a functional target of ALKBH5 in trophoblasts

To further elucidate the ferroptosis-related target gene of ALKBH5, we re-analyzed a published RNA-seq data set (Li et al., 2019). A total of 3,183 genes were identified as dysregulated in response to ALKBH5 knockdown (Fig. 5A). GO and KEGG enrichment analyses of the differentially expressed genes showed that after knockdown of ALKBH 5, cell proliferation and cell death-related genes sets were dysregulated (Fig. 5B). The intersection of the differentially expressed genes after ALKBH5 knockdown and the FerrDB database showed that 47 ferroptosis-related genes were dysregulated after knockdown of ALKBH5 (Fig. 5C). A heat map was generated to show the differential expression of these 47 genes (Fig. 5D). GO and KEGG enrichment analyses were performed for the 47 genes (Fig. 5E). Analysis using the STRING database showed the interactions between genes involved in ferroptosis (Fig. 5F). Heat maps of gene expression were generated for the genes involved in ferroptosis (Fig. 5G). Using oxidative markers and Ferroptosis as keywords, a search was conducted at the GENECARDS website, resulting in 11,435 and 1,407 genes, respectively. Taking the intersection of these genes and the genes that were upregulated and downregulated after knocking down ALKBH5, we showed that 105 of the oxidative markers associated with ferroptosis were downregulated and 75 were upregulated after knocking down ALKBH5 (Fig. 5H). Then, we conducted GO-KEGG analysis using these 105 and 75 genes respectively. This analysis showed that knocking down ALKBH5 resulted in downregulation of 105 genes related to the cell’s ability to respond to oxidative stress (Fig. 5I). However, there were no GO terms and KEGG pathways enriched by the 75 upregulated genes.

Figure 5 The published transcriptomes of ALKBH5 knockdown HTR8 cells and control cells.

(A) The volcano plot showing gene expression changes after ALKBH5 knockdown. (B) GO and KEGG enrichment analysis after ALKBH5 knockdown. (C) Venn diagram showing the intersection of the differentially expressed genes after ALKBH5 knockdown and gene lists from the FerrDB database. (D) Heatmap showing the differential expression levels of the 47 genes in (C). (E) GO and KEGG enrichment analysis performed using the 47 genes in (C). (F) Network analysis of Ferroptosis-related genes in (D) from the String database. (G) Heat map showing ferroptosis-related gene expression levels. (H) Venn diagram showing the intersection of the genes upregulated after knockdown ALKBH5, the genes downregulated after knockdown ALKBH5, the genes related to ‘Oxidative markers’ in the GENECARDS database, and the genes related to ‘Ferroptosis’ in the GENECARDS database. (I) GO and KEGG enrichment analysis for the 105 genes in (H).

Comparing these 105 downregulated genes with the differentially expressed genes from Fig. 1B, we chose FTL as the candidate gene for further research and established HTR8 cells with ALKBH5 knockdown (Figs. S2D–2F). Western blotting showed that knockdown of ALKBH5 decreased the expression of FTL (Figs. 6A, 6B). Consistent with the results of the knockdown experiments, overexpression of ALKBH5 upregulated the expression of FTL (Figs. 6C, 6D). Moreover, RSL3 stimulation also upregulated the expression of FTL (Figs. 6E, 6F).

Figure 6 ALKBH5 regulated the expression of FTL.

(A, B) Western blotting results showing that knockdown ALKBH5 reduced FTL protein levels. Experiments were repeated three times. (C, D) Western blotting results showing that overexpression of ALKBH5 increased FTL protein levels. Experiments were repeated three times. (E, F) Western blotting results showing that RSL3 increased the expression of FTL. Experiments were repeated three times. (G, H) Western blotting results showing that FTL was increased in the villous tissues of patients with RM (n = 14) compared with that in HC (N = 14). (I, J, K) Representative immunohistochemical images depicting the expression of FTL in the villous tissues from patients with RM (n = 10) and HC (n = 10). (CTB, cytotrophoblast; STB, syncytiotrophoblast; * p < 0.05; ** p < 0.01).

Increased FTL in cytotrophoblasts of patients with RM

For further verification of the role of the ALKBH5-FTL axis in the pathogenesis of RM, we examined the FTL expression levels in the villous tissues of patients with RM or the HCs. Western blotting showed that FTL levels were increased in the villi of patients with RM compared with those in the HCs (Figs. 6G, 6H). Immunohistochemical staining also confirmed that the expression of FTL was increased in the cytotrophoblasts of patients with RM (Figs. 6I–6K). In summary, the activity of ALKBH5-FTL axis was likely to be increased in the cytotrophoblasts of patients with RM.

Discussion

The function of cytotrophoblasts is critical for normal placental formation and fetal development, and dysfunction of these cells might lead to miscarriage or spontaneous abortion (Brown et al., 2019). Accumulating evidence confirms that m6A affects programmed cell death, including ferroptosis, in the development of disease (Wu et al., 2024; Xu et al., 2022). However, little is known about m6A regulation of ferroptosis in patients with RM.

In this study, we observed that villous samples from patients with RM had higher amounts of ALKBH5 and FTL than the HCs. In vitro experiments revealed that the ferroptosis of HTR-8 cells was inhibited by ALKBH5 under RSL3 induction. We further discussed the potential mechanism of m6A and ferroptosis in RM. Deciphering the precise and diverse mechanisms in regulating trophoblast cell ferroptosis will offer new insights for RM therapeutic strategies.

The pathogenesis of RM is complex and unclear; hence, the treatment of RM remains difficult, resulting in mental and physical suffering of patients and families (Farren et al., 2021; Murphy, Lipp & Powles, 2012). Despite intense research efforts to elucidate the pathogenesis of RM and seek treatment options, currently, there are no effective treatments for RM (Dimitriadis et al., 2020). Hence, it is imperative to conduct further investigations to gain a deeper understanding of the pathogenesis and pathologies of RM. We observed that the iron content in the villous tissue of patients with RM was higher than that of healthy individuals, accompanied by higher ALKBH5 and FTL expression. ALKBH5 has been reported to modulate ferroptosis (Lv et al., 2023; Ye et al., 2022). Stimulation by RSL3 reduced m6A levels in HTR8 cells, while increasing the expression levels of ALKBH5 and FTL. In addition, ALKBH5 alleviated cell death induced by RSL3 and positively regulated the expression of FTL. These results suggest a potential role of the m6A eraser ALKBH5 in the modulation of ferroptosis in cytotrophoblast dysfunction during RM.

The most common RNA modification is m6A, which is involved in the regulation of RNA splicing, translation, and stability (Bartosovic et al., 2017). Based on our bioinformatic analysis using previously published transcriptome data (Tian et al., 2016) and the quantification of m6A in the total RNA of villous tissues from patients with RM compared with that in the HCs, we chose ALKBH5 for further research. ALKBH5 is an m6A demethylase, which has been shown to play fundamental roles in the onset and development of reproductive disorders. Higher ALKBH5 levels in the villi of patients with RM inhibited the migration and invasion of extravillous trophoblasts (EVTs) by regulating the stability of CYR61 mRNA (encoding cysteine rich angiogenic inducer 61) (Li et al., 2019). Mouse experiments showed that inhibition of ALKBH5 alleviated preeclampsia-like symptoms through the Wnt/β-catenin pathway (Guo, Song & Yang, 2022). In our study, combining the published RNA-seq data of HTR8 cells upon ALKBH5 knockdown, the published RNA-seq data of patients with RM and HC, and the FerrDB database, identified FTL as the candidate gene that is regulated by ALKBH5. Ferritin heavy chain (FTH1) and FLT constitute ferritin, which acts as an iron reservoir responsible for storing and releasing iron (Gatica, Lahiri & Klionsky, 2018). Most studies reported that overexpression of FTL promoted iron storage, thereby inhibiting ferroptosis (Chen et al., 2020; Xie et al., 2016). This is consistent with the overexpression of ALKBH5 functioning to alleviate ferroptosis. It has been reported that the content of Fe2+ at the implantation site of RM mice was higher than that in the control group (Lai et al., 2024). Consistent with this model, our results showed enhanced Prussian blue staining in the villous tissues of the patients with RM. Collectively, these results suggested that ferroptosis occurs in RM. Subsequently, we detected increased ALKBH5 and FTL levels in patients with RM and the ALKBH5′s ferroptosis resistance function. Finally, we verified that the expression of FTL, encoding a protein regulating the storage of iron, was regulated by ALKBH5. Our results showed that the ALKBH5-FTL axis was upregulated during ferroptosis in cytotrophoblasts, and induced resistance to ferroptosis. However, ALKBH5 and FTL expression levels were elevated in the RM group with aggravated ferroptosis, suggesting that the complex biological function of ALKBH5 remains to be explored. There are still limitations to the study. The mechanism by which ALKBH5 regulates the expression of FTL and the function of the ALKBH5-FTL axis in pregnancy outcome are unclear.

Disrupted iron homeostasis accompanied by increased lipid peroxidation and elevated ROS content are characteristics of ferroptosis (Gentric et al., 2019). In the cytotrophoblasts of patients with RM, not only was the iron content increased, but also the expression of ALKBH5 and FTL were augmented. As an immortalized cell line established using early pregnancy cytotrophoblast cells (Jain et al., 2017), HTR8 cells were used to investigate the role of the ALKBH5-FTL axis in the occurrence of ferroptosis in cytotrophoblasts. Under the stimulation by the ferroptosis agonist RSL3, the expression levels of ALKBH5 and FTL were elevated. Overexpression of ALKBH5 resulted in higher cell viability, a reduced GSSG/GSH ratio, decreased ROS levels, and decreased MDA contents in HTR8 cells under RSL3 stimulation compared with cells transfected with the empty plasmid. Analysis of oxidative markers from published data and the GeneCards database showed that knocking down ALKBH5 resulted in overall downregulation of GO terms and KEGG pathways related to oxidative stress. These results confirmed that ALKBH5 alleviates the ferroptosis phenotype of the cytotrophoblasts, which might be a spontaneously self-protection mechanism formed in the cytotrophoblasts of patients with RM.

Collectively, by a comparative analysis of the published tissue RNA-seq data, the ALKBH5 knockdown HTR8 cells RNA-seq data, and the FerrDB database, this study provided new insights into the pathological mechanism of RM by exploring the function of ALKBH5. To study whether ALKBH5-FTL has direct effect on pregnancy outcome, MeRIP-seq and further in vivo experiments are required. The results of this study will increase our understanding of the pathogenesis of RM and provide guidance to predict the onset of RM. Furthermore, the ALKBH5-FTL axis might have potential therapeutic value for the treatment of RM.

Conclusions

In conclusion, the results of the present study suggested that recurrent miscarriage might result, at least in part, from ferroptosis of cytotrophoblasts. ALKBH5 expression was increased in the cytotrophoblasts of patients with RM compared with that in the HCs. Furthermore, ALKBH5 alleviated the changes in ferroptosis-associated oxidative markers induced by RSL3. Finally, we confirmed that ALKBH5 regulated the expression of FTL in cytotrophoblasts. In summary, elevated ALKBH5 in cytotrophoblasts of patients with RM alleviated ferroptosis through regulation of FTL.

Supplemental Information

Supplemental Information 1 Primers used for qPCR in the present study

Supplemental Information 2 Uncropped western blotting images

Supplemental Information 3 Efficiency of the knockout and overexpression of ALKBH5

(A) qRT-PCR results showing ALKBH5 mRNA levels in HTR8 cells transfected with pLV or pLV-ALKBH5. (B, C) Western blotting results showing ALKBH5 levels in HTR8 cells transfected with pLV or pLV-ALKBH5. (D) qRT-PCR results showing ALKBH5 mRNA levels in HTR8 cells transfected with siNC or siALKBH5. (E, F) Western blotting results showing ALKBH5 levels in HTR8 cells transfected with siNC or siALKBH5. (**p < 0.01).

Supplemental Information 4 MIQE checklist

We are grateful to Dr. Fu-Ju Tian for his mentorship and guidance.

Additional Information and Declarations

Competing Interests

Author Contributions

Human Ethics

Data Availability

The authors declare there are no competing interests.

Chuanmei Qin performed the experiments, analyzed the data, prepared figures and/or tables, authored or reviewed drafts of the article, and approved the final draft.

Jiayi Wu performed the experiments, analyzed the data, prepared figures and/or tables, authored or reviewed drafts of the article, and approved the final draft.

Xiaowei Wei performed the experiments, authored or reviewed drafts of the article, and approved the final draft.

Xueqing Liu performed the experiments, authored or reviewed drafts of the article, and approved the final draft.

Yi Lin conceived and designed the experiments, analyzed the data, authored or reviewed drafts of the article, and approved the final draft.

The following information was supplied relating to ethical approvals (i.e., approving body and any reference numbers):

The study was approved by the Medical Ethics Committee of the International Peace Maternity & Child Health Hospital of China Welfare Institute, Shanghai [(GKLW) 2021-49].

The following information was supplied regarding data availability:

The data is available at figshare: Chuanmei, Qin; Wu, Jiayi; Wei, Xiaowei; Liu, Xueqing; Lin, Yi (2024). Elevated ALKBH5 is associated with ferroptosis-related dysfunctions in cytotrophoblasts through FTL in patients with recurrent miscarriage. figshare. Journal contribution. https://doi.org/10.6084/m9.figshare.24902073.v1.

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
