# Peer review of "ALKBH5 modulation of ferroptosis in recurrent miscarriage: implications in cytotrophoblast dysfunction"

_PeerJ, doi:10.7717/peerj.18227_

## Round 0.1 · original submission · Minor Revisions

Dear authors. Thank you for your submission. After reviewing the reviewers' comments, I ask for revisions before the manuscript can be considered further. Please address all reviewer comments to improve the quality of the manuscript. Also, make sure that all corrections are highlighted in a tracked-changes copy of the manuscript before sending it back.

**Language Note:** The review process has identified that the English language must be improved. PeerJ can provide language editing services - please contact us at [email protected] for pricing (be sure to provide your manuscript number and title). Alternatively, you should make your own arrangements to improve the language quality and provide details in your response letter. – PeerJ Staff

·

Basic reporting

1. In general, the manuscript is well written, except for a few sections where I have provided my comments for rephrasing.

2. It requires proofreading in English

3. There are recent references, but certain sections require additional references

Experimental design

1. Authors clearly define the research objective and support it with a well-structured study design

2. During the study, human subjects were used. It is stated that the study has been approved by the ethical committee. Please clarify on termination of the pregnancy. What is the reason for abortion induction?

3. The study parameters include ROS, GSSG and MDA. Please explain why this oxidative markers are not included in the findings and conclusion.

Validity of the findings

1. The data are robust and statistically sound.

2. Conclusion should include oxidative markers.

Additional comments

Ive provided pdf file with my feedbacks

Reviewer 2 ·

Basic reporting

Overall Evaluation: This manuscript presents a compelling investigation into the role of ALKBH5 in ferroptosis and its association with recurrent miscarriage. The study is methodologically sound, with clear relevance to understanding the pathogenesis of recurrent miscarriage. The findings contribute valuable insights into ferroptosis mechanisms in cytotrophoblasts, highlighting potential therapeutic targets. I think it accepting this manuscript for publication after minor revisions.

Experimental design

1. The logical progression from identifying ALKBH5 as a candidate gene to its functional analysis in ferroptosis is well established. However, the discussion could be strengthened by explicitly connecting how these findings fill the identified knowledge gap and their potential implications for therapeutic strategies in recurrent miscarriage.
2. In the cell culture experiments, the conditions under which cells were cultured and treated (e.g., concentrations of RSL3 and duration of treatment) are provided, but the rationale behind choosing these specific conditions is not. Clarifying why these conditions were chosen would strengthen the methodological rigor.
3. The manuscript could benefit from more detailed descriptions of the immunohistochemistry (IHC) and western blotting procedures, particularly in terms of antibody validation. Providing information on how antibody specificity was confirmed would enhance the credibility of the protein expression data

Validity of the findings

1. The usage of Student's t-test or Welch's t-test for data that appears to have a normal distribution is mentioned, but it's not clear how the normality of the data was assessed. I'd suggest stating explicitly the statistical test (e.g., Shapiro-Wilk test) used for verifying the assumption of normality for the dataset.
2. For the analysis involving multiple comparisons, such as the differential gene expression analysis, the lack of mention of adjustment for multiple testing (e.g., Bonferroni, Benjamini-Hochberg) is noted. To enhance the reliability of the findings, I propose including a method for controlling the false discovery rate (FDR).
3. While the manuscript includes figures to represent the data, there's a missed chance to use heatmaps for visualizing the expression levels of differentially expressed genes involved in ferroptosis. I suggest adding heatmaps with hierarchical clustering to better illustrate the gene expression patterns and their relationships.
4. The current title is informative but could be streamlined for clarity and impact. You might consider revising it to "ALKBH5 Modulation of Ferroptosis in Recurrent Miscarriage: Implications in Cytotrophoblast Dysfunction" to more succinctly convey the main findings and significance.
5. In the context of your study on ALKBH5's role in ferroptosis and its implications for recurrent miscarriage, I'd recommend looking into the study titled "m6A regulator-mediated RNA methylation modification patterns are involved in immune microenvironment regulation of periodontitis." This reference might offer valuable insights into the wider implications of RNA methylation modifications, especially m6A regulators like ALKBH5, in altering the immune microenvironment. Considering the considerable overlap in ferroptosis mechanisms and immune regulation, this citation could enrich the discussion of your results by showing a parallel between the pathophysiological processes in recurrent miscarriage and periodontitis. Adding this reference could not only broaden the context of your research but also highlight the complex roles of m6A modifications in various diseases, underlining the relevance of your work to broader biomedical research.

---

## Round 0.2 · accepted · Accept

Dear Authors,

Thank you for your resubmission. I have noticed that you have significantly changed the manuscript according to the reviewers' comments. In addition, the overall quality of the manuscript has been improved, including the English language. Therefore, I fully support the acceptance of this manuscript.

Reviewer 2 ·

Basic reporting

After reviewing the revised manuscript, I am pleased with the significant improvements made. The authors have effectively addressed the previous concerns, enhancing the overall quality and ensuring it meets publication standards. I fully support its publication and look forward to its contribution to our field.

Experimental design

no

Validity of the findings

no